# Spontaneous symmetry breaking in polar fluids

Calum J. Gibb [1], Jordan Hobbs [2], Diana I. Nikolova [2], Thomas Raistrick[2], Stuart R. Berrow [2], Alenka Mertelj [3], Natan Osterman[3,4], Nerea Sebastián [3], Helen F. Gleeson [2] & Richard. J. Mandle [1,2] ✉

Spontaneous symmetry breaking and emergent polar order are each of fundamental importance to a range of scientific disciplines, as well as generating rich phase behaviour in liquid crystals (LCs). Here, we show the union of these phenomena to lead to two previously undiscovered polar liquid states of matter. Both phases have a lamellar structure with an inherent polar ordering of their constituent molecules. The first of these phases is characterised by polar order and a local tilted structure; the tilt direction processes about a helix orthogonal to the layer normal, the period of which is such that we observe selective reflection of light. The second new phase type is anti-ferro-electric, with the constituent molecules aligning orthogonally to the layer normal. This has led us to term the phases the $SmC_P^H$ and $SmA_{AF}$ phases, respectively. Further to this, we obtain room temperature ferroelectric nematic ($N_F$) and $SmC_P^H$ phases via binary mixture formulation of the novel materials described here with a standard $N_F$ compound (DIO), with the resultant materials having melting points (and/or glass transitions) which are significantly below ambient temperature. The new soft matter phase types discovered herein can be considered as electrical analogues of topological structures of magnetic spins in hard matter.

Spontaneous symmetry breaking manifests in a wide range of scientific disciplines and ongoing problems: the Higgs mechanism in subatomic physics[1]; autocatalysis in chemistry[2]; homochirality in biology[3]; and the Dzyaloshinskii–Moriya interaction in ferromagnetism and ferroelectrics leading to the formation of magnetic skyrmions, twisted (chiral) magnetic structures, and other exotic spin textures[4]. A lesser-known subset of materials which spontaneously break symmetry are liquid crystals (LCs). LCs are fluids which can also exhibit spontaneous symmetry breaking through the formation of helical superstructures[5–11] or as a means to escape from local polar ordering in highly ordered systems[12].

LCs are synonymous with LCD technology, and, with this remaining the principal means of information display, the discovery of new liquid crystal phases at equilibrium is typically regarded as being of the highest significance. Current materials exploited in LCDs exhibit nematic (N) phases and are apolar (Fig. 1a), despite being formed from molecules with large electric dipole moments, due to the inversion symmetry of the director, and so the bulk material is polarisable but not polar. Despite being the subject of theoretical interest in the 1920's[13], the lack of experimental discovery condemned polar nematic phases to obscurity for almost a century. The discovery of polar nematic phases at equilibrium in the late 2010's[14,15] has garnered significant excitement and has been described as having "promise to remake nematic science and technology"[16]. This is now referred to as the ferroelectric nematic ($N_F$) phase and is comprised of molecules with large electric dipole moments which align giving rise to a phase bulk polar order[17–20]. The lack of inversion symmetry means that the $N_F$ phase possesses $C_{\infty v}$ symmetry with polarization along the director (n̂)

[1]School of Chemistry, University of Leeds, Leeds, UK. [2]School of Physics and Astronomy, University of Leeds, Leeds, UK. [3]Jožef Stefan Institute, Ljubljana, Slovenia. [4]University of Ljubljana, Faculty of Mathematics and Physics, Ljubljana, Slovenia. ✉e-mail: r.mandle@leeds.ac.uk

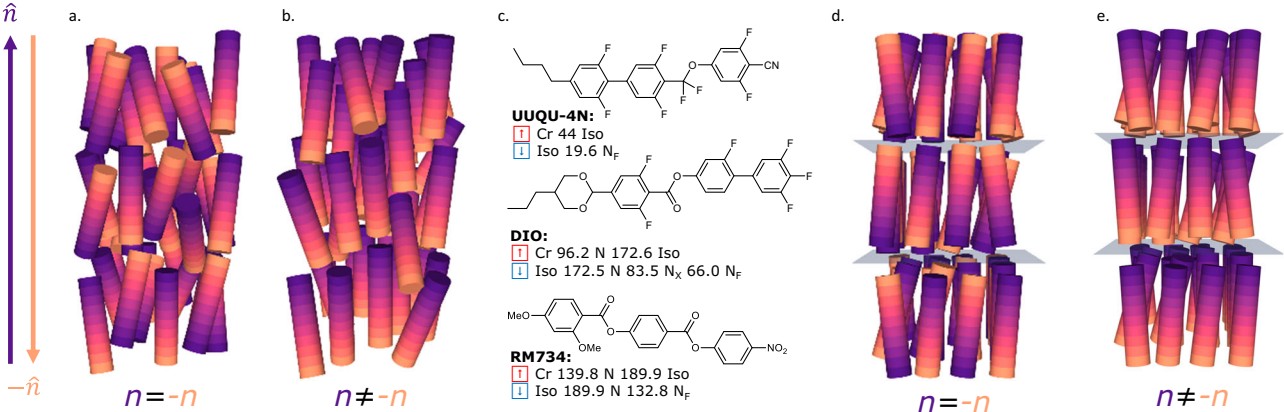

**Fig. 1 | Schematic depictions of polar and apolar liquid crystal phases and materials. a** Apolar nematic; **b** polar nematic; **c** the structures of the exemplar $N_F$ materials: UUQU-4-N[19]; DIO[15]; RM734[43] and, **d** apolar and **e** polar smectic A phases. The lack of inversion symmetry within the polar variants of the N and SmA phases results in both phases possessing $C_{\infty v}$ symmetry with the polarization direction parallel to the director n̂ along the long molecular axes.

(i.e. largely parallel to the long molecular axis). Although in its infancy, the $N_F$ phase has been suggested as candidate for multiple applications including as photo-variable capacitors[20], electrostatic actuators[21] as well as the next generation of display devices. Current materials (for example those in Fig. 1c) present the unwelcome combination of challenging working temperature ranges and often low chemical stability. Currently disclosed materials and mixtures are metastable, even if the $N_F$ phase can be cooled to ambient temperatures it eventually solidifies. This has driven a significant appetite for new materials that can sustain the polar mesophase at and below ambient temperatures[20,22].

Due to the low energy cost of elastic deformation of many liquid crystalline phases, polar order has recently been discovered in complex LC phase types[23–30]. Longitudinal ferroelectric variants of the apolar orthogonal smectic (lamellar) A (SmA) phase (Fig.1d) have recently been reported in both pure materials[31] and in binary mixtures[32]. Molecular rotation orthogonal to the long molecular axis in the ferroelectric SmA ($SmA_F$) phase is essentially frozen, again resulting in a phase with no inversion symmetry ($C_{\infty v}$) and polarization parallel to n̂, along the long molecular axis. Tilted smectic phases (such as SmC phases) can adopt a helical superstructure with polar order when the constituent molecules are chiral (e.g. SmC*[33], as the introduction of chirality reduces the symmetry from $C_h$ to $C_2$ and so leads to a polarisation perpendicular to the layer normal. If the helix is unwound, either through application of a field or surface treatment, a macroscopic polarisation (orders of magnitude smaller than in the $N_F$ or $SmA_F$ phases) results[16,34]. The polar SmC equivalent of the $N_F/SmA_F$ phase has not yet been reported.

Herein we report a family of rod-like LC materials with large electric dipole moments roughly parallel to the long molecular axis. Some members display a polar SmC phase which spontaneously adopts a helical superstructure. Others display an antiferroelectric orthogonal SmA phase. Impressively, simple binary mixtures afford materials that are operable at (and far below) ambient temperatures.

## Results

During initial polarising microscopy studies, we observed compound **1** to exhibit selective reflection of light in a manner reminiscent of chiral nematic phases, although the chemical structure itself is achiral. This prompted the investigations detailed below, with our principal focus here on compounds **1, 4** and binary mixtures of **1** with DIO, however, the phase behaviour of all 4 compounds synthesised are detailed in Table 1. Initial phase assignment for **1** was made by microscopy on cooling from the isotropic liquid (Fig. 2a–c, S1). First, a nematic (N)

phase forms (Fig. S1), identified by its characteristic schlieren texture when viewed between untreated glass slides. Further cooling of the N phase yields two orthogonal smectic phases: at higher temperatures a SmA phase which displays a focal-conic and fan texture (Fig. 2a), and a second smectic phase at lower temperatures in which the focal conics and fans become smooth and small regions appear close to the fan nucleation sites wherein the optical retardation changes rapidly across the sample (Fig. 2b, S2). Measurements of the current response of 1 (Fig. 2d) shoe first smectic phase is apolar. The phase immediately below the apolar SmA phase has a single peak in the current response, indicating the phase is ferroelectric (Fig. 2e). Studies of the temperature dependence of the layer spacing (d(T); vide infra) confirm that the second smectic phase is indeed orthogonal and thus is designated as the $SmA_F$. The layer spacing of the $SmA_F$ phase is essentially temperature independent and of the order of a single molecular length ($\approx$3 nm at the B3LYP-GD3BJ/aug-cc-pVTZ level of DFT) and so is a monolayer smectic phase.

Cooling the $SmA_F$ phase of **1** below −90 °C yields a further phase transition wherein the smooth fan texture becomes significantly disrupted, yielding a texture with many small domains. The disruption quickly subsides allowing the texture to resolve into one similar to that seen in the $SmA_F$ phase, with some striations now visible along the backs of the smooth blocks reminiscent of the SmA-SmC phase transition (Fig. 2c, S2). Current response measurements confirm that polar order is retained upon entering this smectic phase, however, the peak in the current response splits into two non-equal peaks (Fig. 2b). The smaller of the two peaks is seemingly associated with the tilt where the tilt is possibly controllable by applied field (detailed discussed found in the SI section 2.4.). A simple visual inspection of **1** in the lower temperature tilted phase confined in a 5-micron cell under ambient lighting conditions clearly demonstrates its optical activity (Fig. 2g). Quantitative measurement of the wavelength dependence via spectroscopy proved challenging due to the highly scattering texture of the tilted phase, which exists as many small domains and the difficulty of aligning the periodicity responsible for the optical activity, along with the inability to obtain homeotropic alignment for observation along the periodic structure although we include our best efforts in the ESI which does indicate a slight temperature dependence of the wavelength (Fig. S18). However, simply placing a circular polariser between the sample and an observer showed changes in both the intensity and colour of the reflected light, indicating the presence of a chiral superstructure within this tilted phase. Due to its ability to reflect visible light for at least some of the temperature regime of the tilted phase, we expect the periodicity of this structure to be of the order of a few hundreds of nanometres.

**Table 1 | Transition Temperatures (T/°C) and associated enthalpies of transition (ΔH/kJ mol⁻¹) for compounds 1–4 determined by DSC at a heat/cool rate of 10 °C min⁻¹; phase assignments were made on the basis of polarized optical microscopy (POM), X-ray scattering, current-response and further experiments as described in the text**

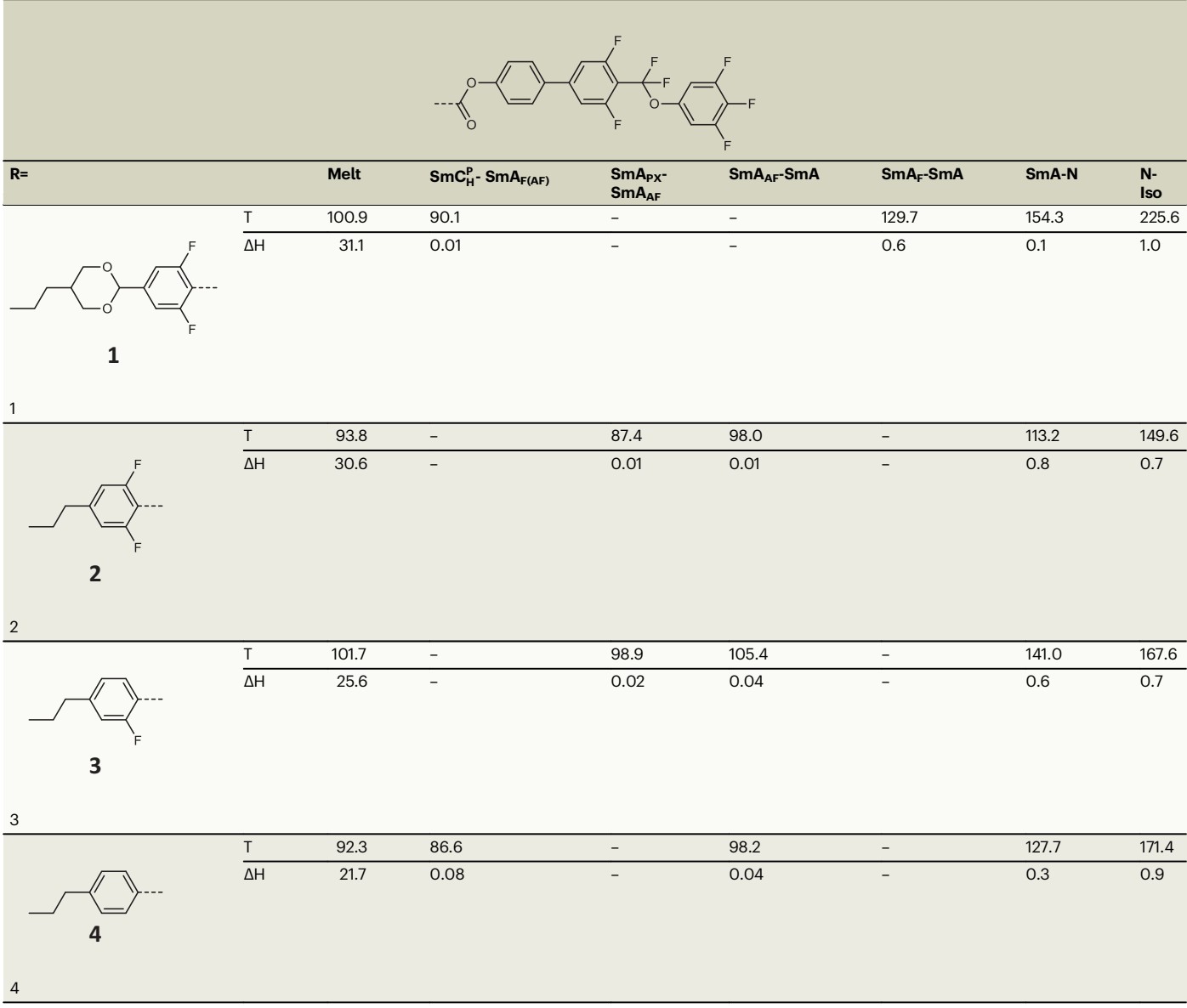

| R= | | Melt | $SmC_H^P$ - $SmA_{F(AF)}$ | $SmA_{PX}$- $SmA_{AF}$ | $SmA_{AF}$-SmA | $SmA_F$-SmA | SmA-N | N-Iso |
|---|---|---|---|---|---|---|---|---|
| **1** | T | 100.9 | 90.1 | – | – | 129.7 | 154.3 | 225.6 |
| | ΔH | 31.1 | 0.01 | – | – | 0.6 | 0.1 | 1.0 |
| **2** | T | 93.8 | – | 87.4 | 98.0 | – | 113.2 | 149.6 |
| | ΔH | 30.6 | – | 0.01 | 0.01 | – | 0.8 | 0.7 |
| **3** | T | 101.7 | – | 98.9 | 105.4 | – | 141.0 | 167.6 |
| | ΔH | 25.6 | – | 0.02 | 0.04 | – | 0.6 | 0.7 |
| **4** | T | 92.3 | 86.6 | – | 98.2 | – | 127.7 | 171.4 |
| | ΔH | 21.7 | 0.08 | – | 0.04 | – | 0.3 | 0.9 |

Square brackets indicate a monotropic phase transition. The $SmA_{PX}$ phase is a 2nd polar orthogonal smectic phase of undetermined structure; see text for discussion.

X-ray scattering measurements on compound **1** confirm the low temperature phase to be tilted by way of a monotonic decrease in layer spacing as the molecules begin to tilt away from the layer normal (Fig. 3a). The growth in tilt is seemingly different to conventional SmC materials[35,36], showing an almost linear temperature dependence and reaching a maximum of 23° at around 30 °C below the SmA-SmC transition, and is not saturated at the point which the material crystallises. We also obtained the tilt angle from the temperature dependence of optical birefringence measurements (Δn) (Fig. 3b), giving values slightly lower than that obtained from X-ray measurements the opposite of what is normally seen for SmC* phases[37].

To gain further insight into the nature of the order within **1**, we elected to measure the 2nd and 4th rank orientational order parameters as a function of temperature via polarised Raman spectroscopy (PRS) (Fig. 3c). In principle, PRS can discriminate between heliconical tilted phases and non-heliconical phases, although careful considerations must be made. For most heliconical structures the measured values of <P2> and <P4> are expected to decrease monotonically as the

heliconical tilt angle grows[38,39], although for some systems a small tilt, coupled with an increase in the order parameter as the lower temperature phase is entered, could balance out such an effect. We find both <P2> and <P4> are approximately constant across the polar and non-polar SmA phases, taking values of −0.8 and −0.5, respectively. On entering the tilted phase there is a decrease in the measured values of <P2> and <P4>, which is consistent with the onset of a heliconical structure. The tilt angle can be inferred from the reduction of <P2> and <P4> on cooling (Fig. S20) suggesting an increasing tilt on cooling consistent with birefringence and X-ray measurements.

To further demonstrate the polar character of the different phases in **1**, Second Harmonic Generation (SHG) investigations were performed in the absence of an applied electric field. No signal is detected either in the N and SmA phases. Cooling into the $SmA_F$ results in a strong SHG signal, which increases upon decreasing the temperature. At the transition to the SmC phase, the SHG signal starts a smooth decrease which is only reverted around 20 degrees below the transition (Fig. 3d). Interestingly, at such temperature selective reflection of

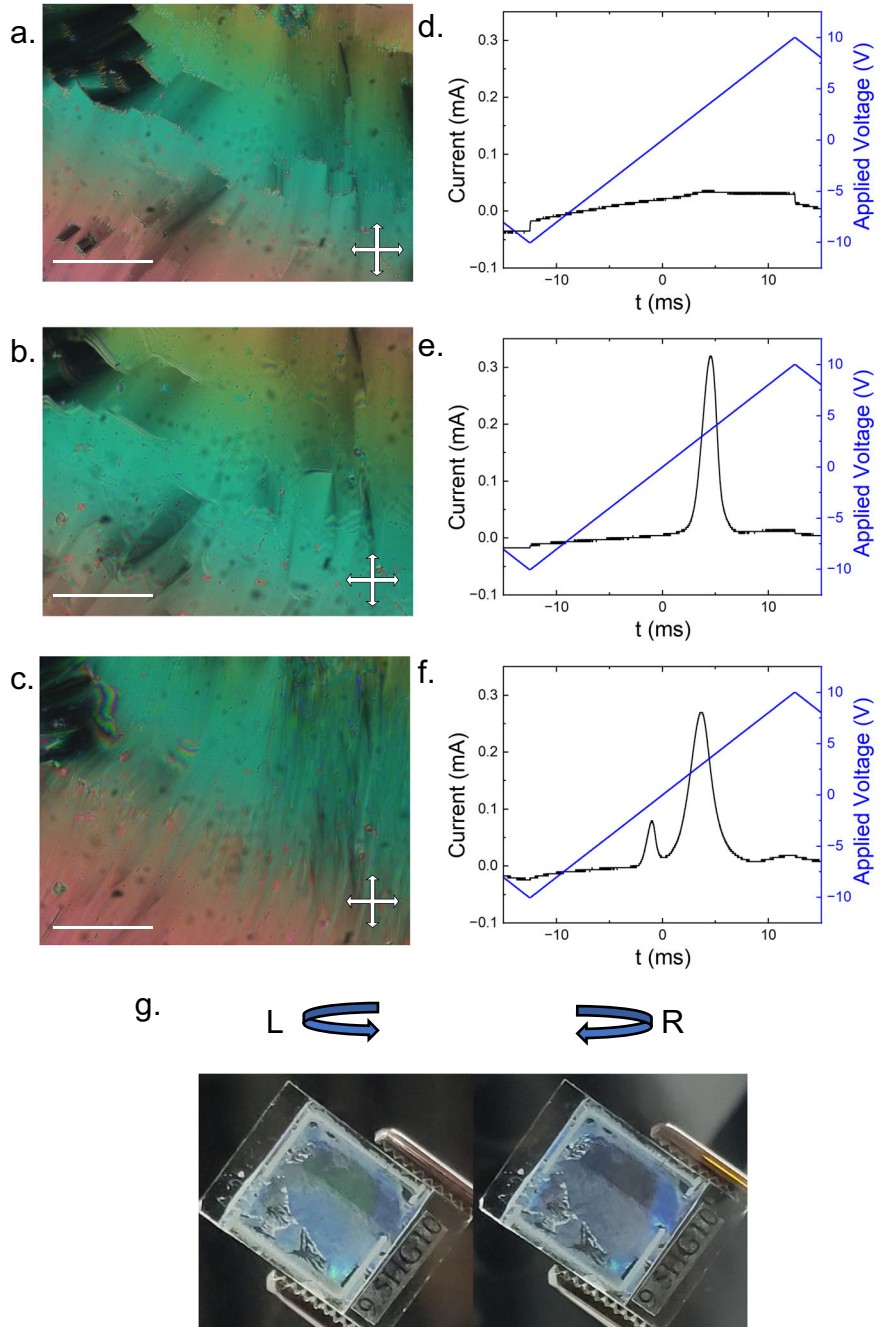

**Fig. 2 | Optical appearance and field response of compound 1.** The SmA phase at 135 °C (**a**); the SmA$_F$ phase at 100 °C (**b**); the SmC$_H^P$ phase at 70 °C (**c**); photomicrographs were captured with the sample between coverslips and under crossed polarisers, with the scale bar in all cases equal to 50 microns. Current response traces measured at 20 Hz (**d**); the SmA phase at 135 °C (**e**); the SmA$_F$ phase at 100 °C (**f**); the SmC$_P^H$ phase at 70 °C (**g**). Photographs of **1** filled into a planar aligned cell taken with a circular polariser of left and right polarisation demonstrating the handedness of the light reflected from the sample; the lack of uniformity across the sample indicates domains of different orientations.

light in the blue region can be observed by simple eye inspection as described above (Fig. S18).

Investigation in parallel rubbed cells for compound **1**, shows the formation of a periodic superstructure in the smectic C phase, with periodicity perpendicular to the rubbing direction. This periodicity slightly evolves on cooling, growing and stabilising at values determined by the confinement thickness, i.e. periodicity is of the order of 2d where d is the cell thickness (Fig. S22). In the thicker cells, two consecutive ribbons exhibit opposite optical activity and SHG interferometric measurements show that polarization alternates from one

ribbon to the next. Both observations combined evidence that chirality and polarization are connected in this system.

In parallel to physical observations, we performed MD simulations of **1** in a polar nematic configuration at a range of temperatures using the GAFF force field in Gromacs 2019.2 (see ESI for full details). Our aim was not to reproduce the periodic structure suggested by experiments to have a pitch of several hundred nanometres, versus the ~9 nm³ volume simulated here, but rather to probe the molecular associations and local phase structure. Each simulation commences from a polar nematic starting configuration, however, **1** readily adopts a lamellar

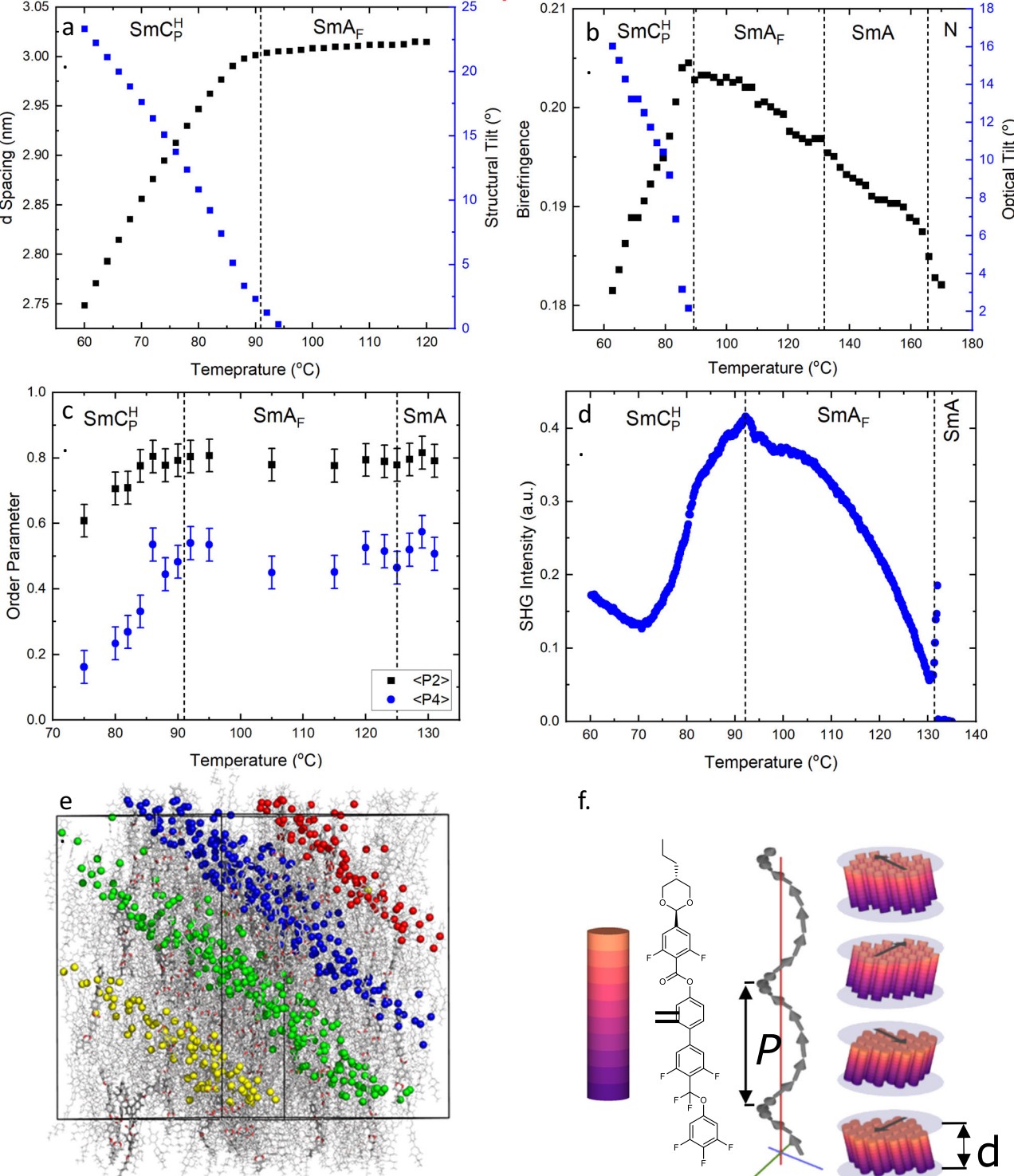

**Fig. 3 | Physical properties of compound 1. a** Temperature dependence of the d-spacing and molecular tilt obtained by SAXS for across the SmA$_F$ and SmC$_P^H$ phases of **1**. **b** Optical birefringence (Δn) and calculated optical tilt angle as a function of temperature for **1**. **c** Temperature dependence of the 2nd and 4th rank orientational order parameters (<P2>, <P4>) obtained by PRS for compound **1**; the magnitude of the error bars result from errors associated with the tolerance of the fitting. **d** Plot of SHG intensity as a function of temperature across the SmA$_F$ and SmC$_P^P$ phases of compound **1**. **e** Instantaneous configuration (t = 249 ns) of a fully atomistic molecular dynamics simulation of **1** in a polar SmC configuration at 400 K. Layers are highlighted by rendering the oxygen atom of each CF$_2$O group as a sphere coloured by layer. **f** The proposed model of the SmC$_P^H$ phase, where P is the helical periodicity (several hundreds of nanometres) and d is the smectic layer spacing (<3 nm, from SAXS). Black arrows are used to illustrate the tilt orientation; the graphic shows four smectic blocks as exemplars, each offset by 90° with a right-handed helical sense, and is not intended to imply discrete clock-like changes.

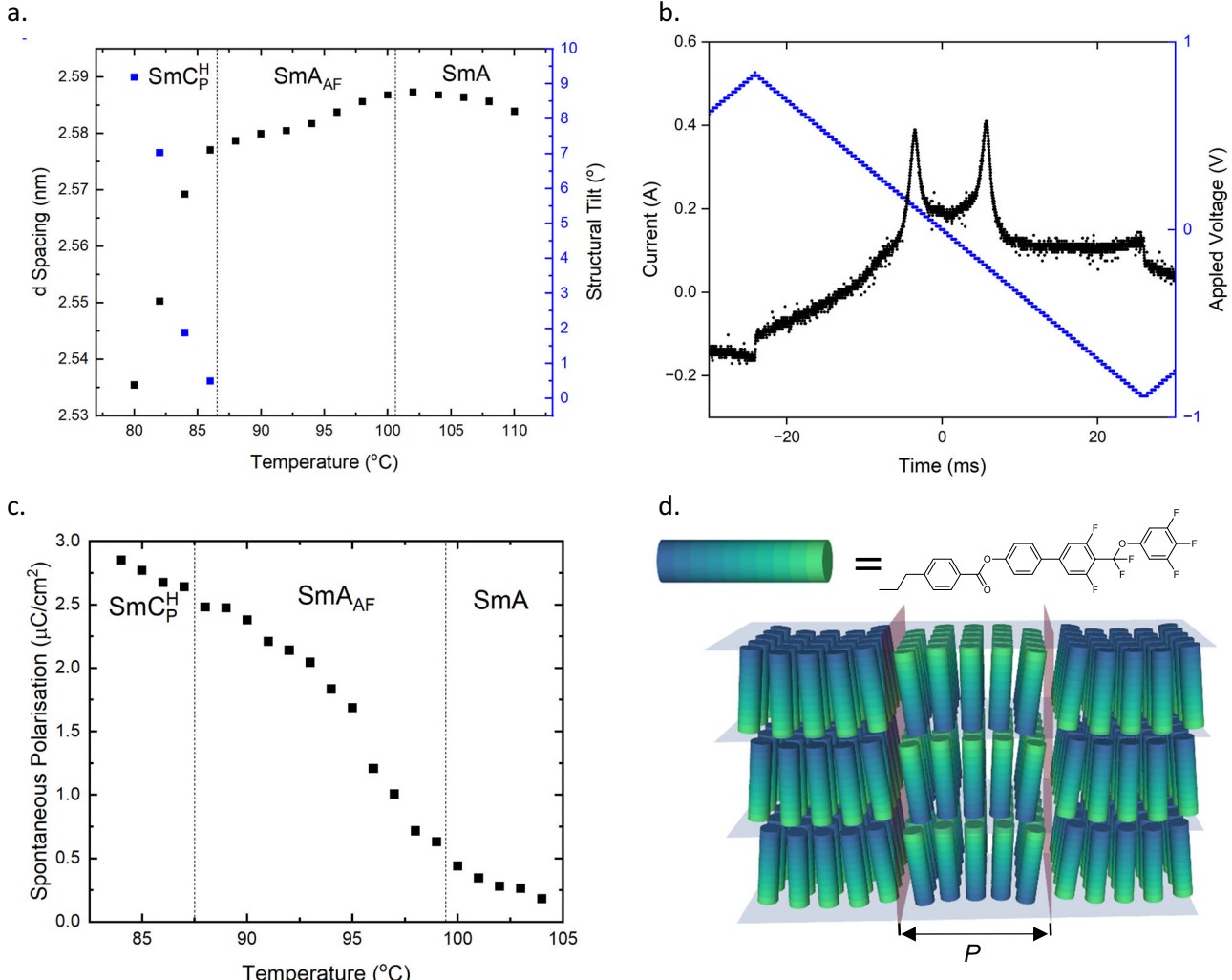

**Fig. 4 | Physical properties of compound 4 and the antiferroelectric smectic A phase. a** Current response at 96 °C in the SmA$_{AF}$ phase measured at 20 Hz; **b** temperature dependence of the spontaneous polarization (P$_s$) measured at 20 Hz in the SmA, SmA$_{AF}$ and SmC$_P^H$ phases; the pre-transitional P$_S$ measured in the SmA is induced polarisation due to the applied voltages analogous to the electroclinic effect at the SmA-SmC* transition; **c** temperature dependence of the d-spacing and tilt angle obtained from SAXS experiments in the SmA, SmA$_{AF}$ and SmC$_P^H$ phases. **d** The proposed model of the antiferroelectric orthogonal smectic A phase (SmA$_{AF}$), where P is the periodicity associated with the polar blocks.

structure (Fig. 3e) and so we observe a polar, tilted phase at temperatures up to 430 K, while at 440 K and 450 K, we observe a polar SmA phase (Fig. S21).

Based on the evidence presented thus far we propose that the lowest temperature phase be denoted as the polar heliconical smectic C (SmC$_P^H$) phase as the phase is a tilted smectic phase with a helical structure and its polar nature is clear from its response to an applied electric field, as well as in SHG studies. We suggest that the simplest structure for such a phase is that shown in Fig. 3f. Such a structure possesses C$_1$ symmetry with the direction of polarization parallel to the layer normal with the spontaneous formation of a helical superstructure, which spontaneously breaks symmetry, likely resulting from the need for the molecules to escape bulk polar order.

As part of our studies into compound **1**, we synthesised a large number of structural analogues, some of which are reported in Table 1. Compounds **2, 3** and **4** exhibit an antiferroelectric response in the second orthogonal SmA phase (Fig. 4a, b) which we therefore denote as SmA$_{AF}$. In agreement with the antiferroelectric character, this phase does not show any SHG signal, as assessed with compound **4**. We are not aware of any prior observation of such a phase of matter. While the textural changes between the SmA and the SmA$_F$ can be quite

significant (Fig. 2, S2), there are few differences between the optical texture of the SmA and SmA$_{AF}$ phases (Figs. S3–5). It has been suggested that polar ordering inhibits the formation of the Dupin cyclides responsible for the fan and focal conic defects found in the natural texture of the (apolar) SmA phase[31]. While we do observe a focal conic texture in the SmA$_{AF}$ phase, this could be due to paramorphosis from the parent SmA phase.

The simplest model of the antiferroelectric SmA phase would be blocks of orthogonal smectic with opposing polar sense, in one sense a lamellar analogue of the Nx[32] (elsewhere referred to as SmZ$_A$, N$_S$ or N$_{AF}$) phase, as shown schematically in Fig. 4d. This structure is likely to experience some splay deformation to minimise free energy. Such a SmA$_{AF}$ phase type was recently suggested based on Onsager–Parsons–Lee local density functional theory[40], although it was also postulated to require possibly unphysical packing fractions of molecules based on steric effects alone. We conjecture that both the SmA$_{AF}$ and SmC$_P^H$ phases have their origins in spontaneous deformation as a means to escape polar order; whereas the tilted smectic phase can readily form a helix, the orthogonal SmA phase would instead form a 1D modulated structure. Many other complex structures can be envisaged, however, and this will be the subject of a future publication.

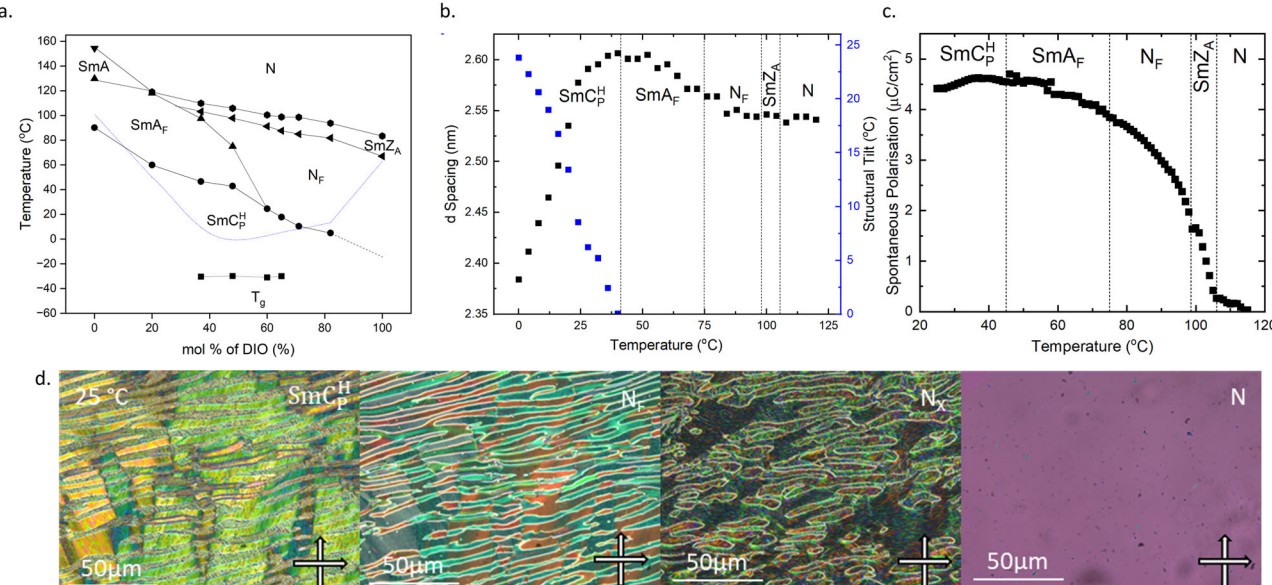

**Fig. 5 | Characterisation of binary mixtures of 1 and DIO. a** Phase diagram of binary mixtures. melting points are denoted by the blue dotted line, and N-I transition temperatures have been omitted for clarity; **b** temperature dependence of the layer spacing (d) for the N, $N_x$, $N_F$, $SmA_F$ and $SmC_P^H$ phases of a mixture of 50% 1 and DIO. For the N, $N_x$ and $N_F$ phase this does not correspond to a layer but corresponds to the long axis correlations present. **c** Temperature dependence of spontaneous polarization ($P_s$) for the same 50% mixture of 1 and DIO measured at 20 Hz, and **d** POM micrographs of a mixture of 35:65 molar ratio of 1 and DIO showing the N, $N_X$, $N_F$ and $SmC_P^H$ phases. The sample is confined within a thin cell treated for planar anchoring.

Additionally, the similarity of the modulation in the $SmA_{AF}$ phase to that in TGB phases[41] suggests a possible source of rich new phase behaviour via expulsion of twist (for example via chirality).

In compound **4**, further cooling of the $SmA_{AF}$ phase yields a transition into a $SmC_P^H$ like phase. Although crystallisation of the sample precludes detailed study, the optical textures (Fig. S5), X-ray scattering patterns (Figs. S10–12) and current response (Figs. S13, 14) are consistent with this assignment. At the $SmA_{AF}$ to $SmC_P^H$ transition, the optical texture changes markedly as the backs of the fans become increasingly striated, much like the transition from SmA to SmC* where the bulk change is molecular tilt and formation of a helix (Fig. S5). For compounds **2** and **3** we observe different behaviour; cooling the $SmA_{AF}$ phase yields a further, seemingly orthogonal smectic phase with polar ordering which we issue the designation $SmA_{PX}$. The optical texture of the $SmA_{PX}$ changes little from the preceding $SmA_{AF}$ phase (Figs. S3, 4), with a vanishingly small associated enthalpy ($\Delta H = 0.01–0.02$ kJ mol$^{-1}$; Fig. S5). There is a slight change observed in SAXS/WAXS, with a reduction in d-spacing and slight increase in FWHM (Figs. S8, 9, S11, 12). The current response (to a triangular wave voltage) in the $SmA_{PX}$ is complex; the symmetric anti-ferroelectric peaks in the $SmA_{AF}$ phase preceding the $SmA_{PX}$ phase become non-symmetric in the lower phase with the peak at negative times having the larger associated area (Fig. S13). This change could indicate a switch to a ferri-electric phase but could equally be explained by some other change affecting the switching behaviour of the materials. The large values of spontaneous polarisation measured for **2** ($-1.7$ μC cm$^2$) and **3** ($-2.7$ μC cm$^2$) evidence the presence of local polar order (Fig. S14), although these are not saturated at the point at which the sample(s) crystallise. While our focus in this communication is introducing the $SmC_P^H$ and $SmA_{AF}$ phases, as well as room temperature polar liquid crystals (vide infra), the observation of further polar fluid phases such as the $SmA_{PX}$ is a reminder of the rich and unexplored physics of these systems.

We envisaged that binary mixtures of **1** with DIO would aid with phase identification through continuous miscibility. Despite this not being the case, the phase diagram (Fig. 5a) displays rich behaviour including polar materials whose melting points and/or vitrification are significantly below 0 °C. Binary mixtures of **1** with concentrations of DIO greater than 50% exhibit N, $N_X$ and $N_F$ phases on cooling, each being identified by a combination of SAXS (Fig. 5b), $P_s$ (Fig. 5c, S16, 17), POM (Fig. 5d) and SHG (Fig. S23). Depending on the composition, we next observe a transition into either the $SmA_F$ or $SmC_P^H$ phase. Notably, a mixture comprising 65% DIO provides the welcome observation of an enantiotropic $N_F$ phase at and below ambient temperatures. For increased concentrations of **1**, the polar nematic phases are not observed and either a N-SmA-$SmA_F$ phase transition is observed. The $SmA_F$ phase then transitions into the $SmC_P^H$ phase which, like the $N_F$ phase in the high concentrations of DIO, is enantiotropic at and below ambient temperature with mixtures comprising 40% DIO exhibiting the most stable example of the phase. The observation of room-temperature $N_F$ and $SmC_P^H$ phases in simple two-component mixtures are particularly exciting as an enabling new materials platform that promises to greatly simplify future experiments and simulations on these remarkable new phases of matter.

It is worth noting that other very recent examples of symmetry breaking in polar fluid systems are coincident with this work. Kumari et al.[42] find that ferroelectric nematics spontaneously adopt a helicoidal (chiral) structure in the absence of anchoring constraints suggesting that this results from an attempt to minimize local polarisation. Additionally, Karcz et al. recently reported a material closely related to compound **1**; this differs from **1** in that it bears an additional fluorine atom on the central benzene ring. Karcz et al. observe a heliconical polar nematic phase (termed $N_{TBF}$) which displays selective reflection of light[11]. These developments point to symmetry breaking by polar ordering being a general phenomenon of relevance to hard-, soft- and bio-materials.

## Methods
### Chemical synthesis
All chemicals and solvents were purchased from commercial suppliers and used as received. Reactions were performed in standard laboratory glassware at ambient temperature and atmosphere and were monitored by TLC with an appropriate eluent and visualised with 254 nm light. Chromatographic purification was performed using a Combiflash NextGen 300+ System (Teledyne Isco) with a silica gel

stationary phase and a hexane/ethyl acetate gradient as the mobile phase, with detection made in the 200–800 nm range. Full details are given in the Supplementary Information.

## Material characterisation

Phase transition temperatures and associated enthalpies of transition for compounds **1**–**4** were determined by differential scanning calorimetry (DSC) using a TA instruments Q2000 heat flux calorimeter at a heat/cool rate of 10 °C min⁻¹, with quoted values given as the average of duplicate runs. Phase identification by polarised optical microscopy (POM) was performed using a Leica DM 2700P polarised optical microscope equipped with a Linkam TMS 92 heating stage for samples were studied sandwiched between two untreated glass coverslips. Birefringence measurements were made by mounting a Berek compensator in this same setup, with the sample confined in 10 µm anti-parallel rubbed planar cells purchased from Instec. X-ray scattering measurements, both small angle (SAXS) and wide angle (WAXS) were recorded using an Anton Paar SAXSpoint 5.0 beamline machine. This was equipped with a primux 100 Cu X-ray source with a 2D EIGER2 R detector. Spontaneous polarisation measurements were undertaken using the current reversal technique, using a Agilent 33220A signal generator and a RIGOL DHO4204 high-resolution oscilloscope, with temperature control via an Instec HCS402 hot stage. Polarized Raman spectroscopy (PRS) was performed using a Renishaw invia Raman spectrometer equipped with a 20 mW 532 nm laser and an optical microscope with a 10× objective. Measurements were performed in well-aligned regions of the sample devoid of defects using an exposure time of 3 × 30 s. SHG investigations were performed using EHC D-type 10 µm thick parallel rubbed cells; see the ESI for full details of the instrumentation used.

## Computational methods

Electronic structure calculations were performed at the B3LYP-GD3BJ/aug-cc-pVTZ level of DFT for a range of low energy conformers. Fully atomistic molecular dynamics (MD) simulations were performed using the General Amber Force Field using RESP charges determined at the B3LYP/6-31G(d) level of DFT. Computational work was performed using the ARC3 and ARC4 computers, part of the high performance computing facilities at the University of Leeds.

## Data availability

The data associated with this paper are openly available from the University of Leeds Data Repository at https://doi.org/10.5518/1488.

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

## Acknowledgements
R. J. M. thanks UKRI for funding via a Future Leaders Fellowship, grant number MR/W006391/1, and the University of Leeds for funding via a University Academic Fellowship. H. F. G acknowledges funding from EPSRC, grant Number EP/V054724/1. The SAXS/WAXS system used in this work was funded by EPSRC via grant number EP/X0348011. RJM and H. F. G gratefully acknowledge support from Merck KGaA. N. S., A. M., and N. O. acknowledge the support of the Slovenian Research Agency (grant numbers P1-0192 and J1-50004).

## Author contributions
C. J. G. and J. H. contributed equally to this work. C. J. G. and R. J. M. performed chemical synthesis; J. H. and C. J. G. performed mixture formulation studies, microscopy, and D. S. C.; J. H. and D. I. N. performed spontaneous polarisation measurements; J. H. performed birefringence measurements, X-ray scattering experiments and selective reflection measurements; T. J. R. performed polarised Raman spectroscopy; S. R. B. Acquired HRMS data and performed method development; N. S., A. M. and N. O. performed S. H. G. studies and microscopy observations in aligned samples; J. H. and R. J. M. performed and evaluated electronic structure calculations; R. J. M. performed and analysed M. D. simulations; H. F. G., A. M. and R. J. M. secured funding. The manuscript was written, reviewed and edited with contributions from all authors.

## Competing interests
The authors declare no competing interests.
