## [Peer Review File · Nature Communications]

REVIEWER COMMENTS

Reviewer #1 (Remarks to the Author):

The manuscript by C. Gibb et al. reports the discovery of different new polar liquid crystalline phases through the combination of polar and/or spontaneous mirror symmetry breaking. The existence of the new phases is supported by different experimental techniques such as polarizing optical microscopy, X-ray diffraction, polarization reversal current measurements, DFT and birefringence measurements.

The results of which consistently lead to conclusion of new polar phases which will open the door for interesting future investigations of the reported materials and other related derivatives.

The authors also successfully show the possibility of getting stable ferroelectric nematic phase at room temperature by mixing compound 1 with the previously reported material DIO.

This is indeed an interesting work which could have an impact on the general understanding of symmetry breaking mechanisms in soft matter. From the scientific point of view, I thus have no objections to see this paper published after the authors considered the points below. I wonder however whether the paper is well-placed as a Communication. Some important results have been transferred to the Supplementary Information and the paper refers to this information. The readability of the paper would be highly improved if this information become part of the paper itself.

Some other points:

1. Please provide HRMS data for the final materials (1-4).
2. The authors should show the DSC thermograms of the new materials is the SI.
3. The Schlieren texture of the N phase in Fig. 2a is not clear. Could the authors provide better texture?
4. The paper deals with mirror symmetry breaking in soft matter and the authors already give references to different systems but they might like to add the following references for additional systems:

Angew. Chem. Int. Ed. 2014, 53, 13115; J. Am. Chem. Soc., 2022, 144, 6936.

5. There are few typing mistakes in the main text and in the SI.

Reviewer #2 (Remarks to the Author):

The evaluated manuscript entitled: 'Spontaneous Symmetry Breaking in Polar Fluids' shows the physical properties of new ferroelectric liquid crystal phases. The authors analyzed four new achiral compounds with longitudinal dipole moments. According to them, compounds 1 and 4 exhibit a previously unknown SmCHP phase, which exhibits the Bragg reflection of light in the visible spectrum. The manuscript presents a hot topic with a huge influence and enormous interest in the liquid crystal community. However, the article needs to be rewritten to make it more transparent and more understandable to the reader. The manuscript may be published in Nature Communications after major revision, in accordance with the suggestions and comments below.

1. We suggest rewriting the manuscript and focusing on examining this effect in compound 1 more broadly. The effect of the formation of the helicoidal structure is not investigated at all. It is pointless to show the behavior of compounds 2, 3, and 4 when we want to explain the behavior of compound 1.

2. The authors cannot discuss the results from compounds 2 and 3, which describe new polar liquid crystal phases, in the Supplementary materials.

3. The Authors should discuss their results in the context of known results (<https://arxiv.org/abs/2311.18552>, Spontaneous polar and chiral symmetry breaking in ordered fluids - heliconical ferroelectric nematic phases by Karcz et al.) which also discussed selective reflection in high polar compounds similar to compound 1.

4. Table 1 contains designations of liquid crystal phases that are not explained in the main text. The manuscript should focus on the phase that exhibits selective reflection.

5. Why do authors observe two polarization peaks in SmCHP? The phenomenon was not explained or discussed.

6. Is the selective reflection observed in SmCHP sensitive to the electric field? Can we tune the selective reflection using the applied electric field?
7. Authors should measure the wavelength of selective reflection. The experiment could use a spectrometer and polarised microscopy in the reflection mode.
8. It would be interesting to know the temperature dependence of selective reflection in the suggested SmCHP phase and its sensitivity to the electric field.
9. Figures are not readable (e.g., Fig 2 a, b).
10. Why does the polarization current in Figure 2b not exceed zero, as shown in Figure 3b?
11. Without an applied electric field, why does the polarization current show a peak in Figure 3b?
12. Authors should perform dielectric spectroscopy measurements of investigated materials. Knowing the relation of electric permittivity in the ferroelectric and antiferroelectric liquid crystal phases will be interesting.
13. Authors should provide the DSC results of the compounds in the supplementary materials.
14. Why only one phase per compound was shown in Table S2?
15. Why is there no difference between the texture photos (SmA-SmAAF-SmCPH) in Figure S2? Although Figure S1 clearly shows the changes in the textures of these phases?
16. What is Q (A-1) parameter in Figure S7?

Reviewer #3 (Remarks to the Author):

The evaluated manuscript entitled: 'Spontaneous Symmetry Breaking in Polar Fluids' shows the physical properties of new ferroelectric liquid crystal phases. The authors analyzed four new achiral compounds with longitudinal dipole moments. According to them, compounds 1 and 4 exhibit a previously unknown SmCHP phase, which exhibits the Bragg reflection of light in the visible spectrum. The manuscript presents a hot topic with a huge influence and enormous interest in the liquid crystal community. However, the article needs to be rewritten to make it more transparent and more understandable to the reader. The manuscript may be published in Nature Communications after major revision, in accordance with the suggestions and comments below.

1. We suggest rewriting the manuscript and focusing on examining this effect in compound 1 more broadly. The effect of the formation of the helicoidal structure is not investigated at all. It is pointless to show the behavior of compounds 2, 3, and 4 when we want to explain the behavior of compound 1.

2. The authors cannot discuss the results from compounds 2 and 3, which describe new polar liquid crystal phases, in the Supplementary materials.

3. The Authors should discuss their results in the context of known results (<https://arxiv.org/abs/2311.18552>, Spontaneous polar and chiral symmetry breaking in ordered fluids - heliconical ferroelectric nematic phases by Karcz et al.) which also discussed selective reflection in high polar compounds similar to compound 1.

4. Table 1 contains designations of liquid crystal phases that are not explained in the main text. The manuscript should focus on the phase that exhibits selective reflection.

5. Why do authors observe two polarization peaks in SmCHP? The phenomenon was not explained or discussed.

6. Is the selective reflection observed in SmCHP sensitive to the electric field? Can we tune the selective reflection using the applied electric field?

7. Authors should measure the wavelength of selective reflection. The experiment could use a spectrometer and polarised microscopy in the reflection mode.

8. It would be interesting to know the temperature dependence of selective reflection in the suggested SmCHP phase and its sensitivity to the electric field.

9. Figures are not readable (e.g., Fig 2 a, b).

10. Why does the polarization current in Figure 2b not exceed zero, as shown in Figure 3b?

11. Without an applied electric field, why does the polarization current show a peak in Figure 3b?

12. Authors should perform dielectric spectroscopy measurements of investigated materials. Knowing the relation of electric permittivity in the ferroelectric and antiferroelectric liquid crystal phases will be interesting.

13. Authors should provide the DSC results of the compounds in the supplementary materials.

14. Why only one phase per compound was shown in Table S2?

15. Why is there no difference between the texture photos (SmA-SmAAF-SmCPH) in Figure S2? Although Figure S1 clearly shows the changes in the textures of these phases?

16. What is Q (A-1) parameter in Figure S7?

Dr. Richard Mandle
Sir William Henry Bragg Building
University of Leeds
Leeds, UK, LS2 9JT

RE: "Spontaneous Symmetry Breaking in Polar Fluids"
NCOMMS-24-14957-T

We thank the editorial staff and the referees for the prompt review of our submission; in light of the referees comments I offer the following response and a revised manuscript on behalf of the authors listed in the revised submission.

We have gathered additional data (second harmonic generation, high resolution mass spectrometry) to support our study and on this basis we include Dr. Nerea Sebastian, Dr. Natan Osterman, Dr. Alenka Mertelj and Dr. Stuart Berrow as co-authors to reflect their contribution to this revised manuscript.

The referee's comments will be presented in indented italics, with our response (and any new or modified figures) in normal text.

REVIEWER COMMENTS

Reviewer #1 (Remarks to the Author):

The manuscript by C. Gibb et al. reports the discovery of different new polar liquid crystalline phases through the combination of polar and/or spontaneous mirror symmetry breaking. The existence of the new phases is supported by different experimental techniques such as polarizing optical microscopy, X-ray diffraction, polarization reversal current measurements, DFT and birefringence measurements.

The results of which consistently lead to conclusion of new polar phases which will open the door for interesting future investigations of the reported materials and other related derivatives. The authors also successfully show the possibility of getting stable ferroelectric nematic phase at room temperature by mixing compound 1 with the previously reported material DIO.

We thank the referee for these comments.

This is indeed an interesting work which could have an impact on the general understanding of symmetry breaking mechanisms in soft matter. From the scientific point of view, I thus have no objections to see this paper published after the authors considered the points below. I wonder however whether the paper is well-placed as a Communication. Some important results have been transferred to the Supplementary Information and the paper refers to this information. The readability of the paper would be highly improved if this information become part of the paper itself.

We are minded to agree. I think the quality of the results here mandates that we elaborate a little more in the main text by moving (and rewriting) some information from the SI. Firstly, we moved discussion around the optical texture of the SmA_AF phase from the SI to the manuscript:

<<NEW TEXT>>

“While the textural changes between the SmA and the SmA_F can be quite significant (Fig 2), there are few differences between the optical texture of the SmA and SmA_{AF} phases (Fig. S4). It has been suggested that polar ordering inhibits the formation of the Dupin cyclides responsible for the fan and focal conic defects found in the natural texture of the (apolar) SmA phase [30]. While we observe a focal conic texture in the SmA_{AF} phase, this could result from to paramorphosis from the parent SmA phase.”

<<END NEW TEXT>>

We also moved discussion of the SmA_{PX} phase to the manuscript:

<<NEW TEXT>>

“For compounds **2** and **3** we observe different behaviour; cooling the SmA_{AF} phase yields a further orthogonal smectic phase with polar ordering which we issue the designation SmA_{PX}. The optical texture of the SmA_{PX} changes little from the preceding SmA_{AF} phase (Fig. S2, S3), with a vanishingly small associated enthalpy ($\Delta H = 0.01-0.02 \text{ kJ mol}^{-1}$; Fig. S5). There is a slight change observed in SAXS/WAXS, with a reduction in d-spacing and slight increase in FWHM (Fig. S7, S8, S10, S11). The current response (to a triangular wave voltage) in the SmA_{PX} is complex; the symmetric anti-ferroelectric peaks in the SmA_{AF} phase preceding the SmA_{PX} phase become non-symmetric in the lower phase with the peak at negative times having the larger associated area (Fig. S12). This change could indicate a switch to a ferroelectric phase but could equally be explained by some other change affecting the switching behaviour of the materials. The large values of spontaneous polarisation measured for **2** ($\sim 1.7 \mu\text{C cm}^{-2}$) and **3** ($\sim 2.7 \mu\text{C cm}^{-2}$) evidence the presence of local polar order (Fig. S13), although these are not saturated at the point at which the sample(s) crystallise. While our focus in this communication is introducing the SmC_P^H and SmA_{AF} phases, as well as room temperature polar liquid crystals (*vide infra*), the observation of further polar fluid phases such as the SmA_{PX} is a reminder of the rich and unexplored physics of these systems.”

<<END NEW TEXT>>

With this change the manuscript is more readable (less cross referencing to the SI is required). We also feel that changes to figures have contributed to improved readability of this work.

Some other points:

1. Please provide HRMS data for the final materials (1-4).

We apologise for the oversight here; we acquired high resolution mass spectrometry data for compounds **1-4**; the acquired data is in accordance with the proposed structures. The method development and data acquisition was performed by Dr. Stuart Berrow who has been included as a co-author on this revision on this basis.

2. The authors should show the DSC thermograms of the new materials in the SI.

DSC thermograms for all compounds have been included in the SI as Figure S5:

Fig. S6: DSC cycles for the first cooling scan after heating into the nematic phase. (a) 1, (b) 2, (c) 3, and (d) 4. For also cycles exothermic heat flow is in the upwards direction.

3. The Schlieren texture of the N phase in Fig. 2a is not clear. Could the authors provide better texture?

We agree with this; an alternative image has been included (in the SI, Figure S1; see below). On reflection we feel Figure 2 in the original submission had too many display elements, being cluttered and therefore not easy to read (small font sizes etc.). We therefore split this figure so that the initial characterisation (microscopy, applied fields) is separated into the 'new' Figure 2 (we have also moved the nematic texture to the SI):

<<New Figure S1>>

Fig. S1: N phase of compound 1 between untreated coverslips. Scale bar indicates 25 μm

<< New Figure 2>>

g. L \rightleftharpoons R

And the in-depth characterisation (SAXS, birefringence, order parameters, SHG, MD-simulations) are included in a 'new' Figure 3:

<<New Figure 3>>

All subsequent figures have been renumbered.

4. The paper deals with mirror symmetry breaking in soft matter and the authors already give references to different systems but they might like to add the following references for additional systems:
Angew. Chem. Int. Ed. 2014, 53, 13115; *J. Am. Chem. Soc.*, 2022, 144, 6936.

We thank the referee for these suggestions and we have included them in the manuscript.

5. There are few typing mistakes in the main text and in the SI.

We have reviewed the manuscript text and SI and corrected typos and grammatical errors as found. For the sake of brevity we do not reproduce a list here.

Reviewer #2 (Remarks to the Author):

Reviewers #2 and #3 have submitted identical reviews, and we will therefore address both here:

The evaluated manuscript entitled: ‘Spontaneous Symmetry Breaking in Polar Fluids’ shows the physical properties of new ferroelectric liquid crystal phases. The authors analyzed four new achiral compounds with longitudinal dipole moments. According to them, compounds 1 and 4 exhibit a previously unknown SmCHP phase, which exhibits the Bragg reflection of light in the visible spectrum. The manuscript presents a hot topic with a huge influence and enormous interest in the liquid crystal community. However, the article needs to be rewritten to make it more transparent and more understandable to the reader. The manuscript may be published in Nature Communications after major revision, in accordance with the suggestions and comments below.

We thank the referee(s) for these supportive comments, and we believe the changes made (detailed herein) are satisfactory.

1. We suggest rewriting the manuscript and focusing on examining this effect in compound 1 more broadly. The effect of the formation of the helicoidal structure is not investigated at all.

We cannot agree to this comment. Contrary to the referee(s) comments the effect of the formation of the helicoidal structure is studied extensively through X-ray, birefringence, polarisation, and microscopy. In this revision we also include new SHG data. Of course there are probably other experiments that could be done, but this comment could be levelled at any scientific paper ever written.

It is pointless to show the behavior of compounds 2, 3, and 4 when we want to explain the behavior of compound 1.

It is disappointing to see the referee(s) describe results as “pointless” and to suggest that we should fragment our work in this way. Compound **4** also exhibits a SmCPH phase as **1** which the referee has perhaps overlooked, generality across different compounds being especially important. The submitted paper details new phases of matter beyond the aforementioned helical phase, and also delivers on the long standing goal of a simple (binary) room temperature N_F system.

2. The authors cannot discuss the results from compounds 2 and 3, which describe new polar liquid crystal phases, in the Supplementary materials.

We agree with the referees comments, in hindsight this was a mistake (albeit only a single paragraph of discussion) and should have been in the main text of the manuscript. This was addressed in response to referee #1 above, and we direct the reader to the response above to reviewer 1.

3. The Authors should discuss their results in the context of known results (<https://arxiv.org/abs/2311.18552>, Spontaneous polar and chiral symmetry breaking in ordered fluids - heliconical ferroelectric nematic phases by Karcz et al.) which also discussed selective

reflection in high polar compounds similar to compound 1.

We agree we can discuss this Karcz compound (which is really interesting, and now published in Science) better; again, this points to the generality of spontaneous symmetry breaking and heliconical phase-types in these materials (Karcz compound, and here **1** and **4**), as does the recent paper of Kumari *et al.* We have added some discussion:

<<NEW TEXT>>

"It is worth noting that other very recent examples of symmetry breaking in polar fluid systems are coincident with this work. Kumari *et al.* [43] find that ferroelectric nematics spontaneously adopt a helicoidal (chiral) structure in the absence of anchoring constraints suggesting that this results from an attempt to minimize local polarisation. Additionally, Karcz *et al.* recently reported a material closely related to compound **1**; this differs from **1** in that it bears an additional fluorine atom on the central benzene ring. Karcz *et al.* observe a heliconical polar nematic phase (termed N_{TBF}) which displays selective reflection of light [10]. These developments point to symmetry breaking by polar ordering being a general phenomenon of relevance to hard-, soft- and bio-materials."

<<END NEW TEXT>>

4. Table 1 contains designations of liquid crystal phases that are not explained in the main text. The manuscript should focus on the phase that exhibits selective reflection.

This largely duplicates points (1) and (2) which have already been addressed. We are pleased the referee(s) enjoyed our discovery of the selective reflection phase, however this is just one of the impact points of this paper (also SmA_AF phase, room temperature N_F and SmCPH phases...). While we appreciate the suggestion and enthusiasm for our discovery, we cannot agree to fragment the paper in this way as this would rob this result of the wider context in which it was generated.

5. Why do authors observe two polarization peaks in SmCHP? The phenomenon was not explained or discussed.

This was discussed with some depth in the SI however we do accept that we did not reference this in the main text in a sufficient way to make this clear to the readers. We suggest that this second peak in the SmCHP phase is due to reformation of the tilted structure which is removed by application of electric field.

In the SI we have added some further analysis to support this in the form of showing strong correlation between growth area of the small peak and growth of molecular tilt. Although this is not conclusive it is at the very least suggestive:

<<NEW TEXT IN SI>>

The smaller peak appears to be associated with the tilt, **fig. S14a** and **S14b** demonstrate the strong correlation between the tilt angle (from X-ray data) and the Ps associated with the small peak. Conversely, the Ps contribution of the larger peak stagnates at the phase transition from the SmA_F phase (~90°C) and does not increase beyond a value of ~3.5mCcm⁻². Although there are many possible contributions to Ps curves known for ferroelectric liquid crystals constrained in devices (see [REF - The Handbook of Liquid Crystals Vol. 4

Smectic and Columnar Liquid Crystals]), this strong correlation leads us to speculate that the initial peak is due to the tilt reforming as the applied voltage is reduced.

Fig. S14b demonstrates the switching mechanism. In the initial part of the curve, where the field is high strength and negative polarity, the system is effectively in the SmA_F configuration. As the field is reduced to lower strength but still negative polarity, the tilt reforms giving back the SmC_P^H structure. Upon increasing the field to cross the switching threshold in the positive polarity domain the polarisation of the material fully switches to match the field resulting in the SmA_F structure again. Such a field-induced reorganisation is necessarily identical to field-induced helical unwinding in this system. It is possible that the introduction of a helical structure is the cause of the “escape from polar order”. Further increasing the polarisation at zero field may cost too much free-energy and so a significant change in the phase structure (i.e. formation of a helix) is cheaper and thus preferable to increasing the longitudinal Ps.

Fig. S15: (a) Measured polarisation for **1**. “Large” and “small” peaks indicate the larger and smaller peak in the current response for material **1** in the SmC_P^H phase. The burgundy line is a guide for the eye and not the result from any fitting. (b) Measured polarisation of the small peak and tilt obtained via X-ray scattering showing the strong correlation between the data. (c) mechanism for polarisation switching for this material as described in the text. The dark tips of the “molecules” indicates the direction of polarisation.

<<NEW TEXT ENDS>>

6. Is the selective reflection observed in $SmCHP$ sensitive to the electric field? Can we tune the selective reflection using the applied electric field?

As stated in the original submission it is not possible to spectroscopically measure the selective reflection due to the highly scattering texture of the phase, which precludes direct measurement.

7. Authors should measure the wavelength of selective reflection. The experiment could use a spectrometer and polarised microscopy in the reflection mode.

We conducted these measurements before the initial submission but were unsatisfied with them due to the lack of homeotropic alignment preventing proper observation along the helix. Along with the significant scattering of the phase SmCHP phase, only a broad structureless peak with no Bragg peaks was observed. We have now included these measurements in the SI to demonstrate the experimental difficulties with this technique currently. Refining this data and measuring this as a function of field (as the reviewer mentions above) would however be a most attractive area for further study.

<<NEW TEXT IN SI>>

Measuring the selective reflection of **1** using a spectrometer is extremely difficult. Selective reflection is observed when the incoming light is parallel to the direction of the periodicity and as such to measure selective reflection homeotropic alignment is required which was found to be unobtainable using polyimide SE1211 (AWAT) or bare ITO electrodes (Instec). However, while good planar alignment was obtained for the N, SmA and SmA_F phase, the SmC_P^H is not uniformly aligned (fig. **S22**, figure in the SHG section) and so part of the incoming light can get selectively reflected before it is then scattered on the defects and irregularities of the sample, which is seen as the defects appearing bluish. This scattered light is the one detected in our setup. The results in fig. **S18** indicate that the pitch length decreases with decreasing temperature before saturating at some amount.

Fig. S18: (a) transmission spectra of **1** in a parallel planar aligned cell. (b) Peak position from transmission measurements. (c) unpolarised images of the electrode area of the cell usage in the spectroscopic measurements. Above 83 °C the wavelength of the reflection is not clear and is generally seen as an increase in scattering which translates to decreased transmission.

<<NEW TEXT ENDS>>

8. It would be interesting to know the temperature dependence of selective reflection in the suggested SmCHP phase and its sensitivity to the electric field.

We agree and this point is mostly addressed in our response to (7); Due to the limitations of the measurements, we felt that further experiments using field could mislead and so we feel this should be revisited once some of the experimental difficulties have been resolved.

9. Figures are not readable (e.g., Fig 2 a, b).

Agreed – as with our response to referee #1 we have changed Figure 2 so that it is split into two figures to aid readability. I direct the referee(s) to our response to point 3 of referee #1.

10. Why does the polarization current in Figure 2b not exceed zero, as shown in Figure 3b?

We thank the referees for noticing this. The measured current had some vertical offset applied which had not been subtracted properly. This does not affect any of the data or analyse other than the position on the y-axis which has been properly corrected. We also noticed that the applied voltage was incorrect by a factor of -1 which has also been corrected.

11. Without an applied electric field, why does the polarization current show a peak in Figure 3b?

Having polarisation peaks equal in area either side of applied voltage polarity reversal is the definition of an anti-ferroelectric material. This is found for the Nx / SmZa phase also, albeit for the Nx phase there is significant conductivity effects making deconvolution of the symmetric peaks more difficult.

12. Authors should perform dielectric spectroscopy measurements of investigated materials. Knowing the relation of electric permittivity in the ferroelectric and antiferroelectric liquid crystal phases will be interesting.

While we agree this would be most interesting, it goes beyond the scope of this current work.

13. Authors should provide the DSC results of the compounds in the supplementary materials.

Absolutely, we apologise for the oversight and - as with our response for referee #1, we have provided DSC thermograms in the SI.

14. Why only one phase per compound was shown in Table S2?

The MD simulations were performed at a single temperature for each compound studied in order to facilitate comparison of their cylindrical distribution functions. The actual phase types here are not especially relevant to this analysis, but regardless we should specify the specific phase formed. Additionally, it is rather spectacular that we form (polar) nematic, SmA, SmC phases. Of course if we varied the simulation temperatures we would expect to generate several different phases, and we did this for compound 1, but to do this for each material would not be especially instructive and goes beyond the scope of this paper.

15. Why is there no difference between the texture photos (SmA-SmAAF-SmCPH) in Figure S2? Although Figure S1 clearly shows the changes in the textures of these phases?

Simply, paramorphosis: the SmCPH phase in Figure S2 (now figure S5) is preceded by a SmA_{AF} phase, whereas in Figure S1 (now figure S2) it is preceded by a SmA_F phase. In each case the observed texture of the SmCPH phase is paramorphotic of the phase above it. The optical textures of the SmCPH phase are not especially diagnostic, yet we caution that these are possibly not the natural textures and are paramorphotic of the preceding mesophase. Again, this points to the importance of generality across materials (and the inclusion of compound 4) which the referee(s) had suggested we remove.

16. What is Q (A-1) parameter in Figure S7?

Q is the scattering vector, $Q = 2\pi/d$

Once again, we would like to thank you for the review of our manuscript and we look forward to hearing the outcome of this revised submission.

Richard J Mandle

Dr. Richard Mandle

UKRI Future Leaders Fellow

@: r.mandle@leeds.ac.uk

REVIEWERS' COMMENTS

Reviewer #1 (Remarks to the Author):

The authors have addressed all my comments and now the paper is clearer for the readers.

The paper could be accepted now for publication in its current form.

Reviewer #2 (Remarks to the Author):

I want to thank the authors for their helpful answers and explanations. I recommend the manuscript for publication in Nature Communications in its current form.

Reviewer #3 (Remarks to the Author):

I want to thank the authors for their helpful answers and all the explanations. I recommend the manuscript for publication in Nature Communications in the present form.